# Intermittency in the *not-so-smooth* elastic turbulence

Rahul K. Singh [1], Prasad Perlekar[2], Dhrubaditya Mitra[3] & Marco E. Rosti [1] ✉

Elastic turbulence is the chaotic fluid motion resulting from elastic instabilities due to the addition of polymers in small concentrations at very small Reynolds (Re) numbers. Our direct numerical simulations show that elastic turbulence, though a low Re phenomenon, has more in common with classical, Newtonian turbulence than previously thought. In particular, we find power-law spectra for kinetic energy $E(k) \sim k^{-4}$ and polymeric energy $E_p(k) \sim k^{-3/2}$, independent of the Deborah (De) number. This is further supported by calculation of scale-by-scale energy budget which shows a balance between the viscous term and the polymeric term in the momentum equation. In real space, as expected, the velocity field is smooth, i.e., the velocity difference across a length scale $r$, $\delta u \sim r$ but, crucially, with a non-trivial sub-leading contribution $r^{3/2}$ which we extract by using the second difference of velocity. The structure functions of second difference of velocity up to order 6 show clear evidence of intermittency/ multifractality. We provide additional evidence in support of this intermittent nature by calculating moments of rate of dissipation of kinetic energy averaged over a ball of radius $r$, $\varepsilon_r$, from which we compute the multifractal spectrum.

Turbulence is a state of irregular, chaotic, and unpredictable fluid motion at very high Reynolds numbers (Re), which is the ratio of typical inertial forces over typical viscous forces in a fluid. It remains one of the last unsolved problems in classical physics. Conceptually, the fundamental problem of turbulence shows up in the simplest setting of statistically stationary, homogeneous, and isotropic turbulent (HIT) flows: What are the statistical properties of velocity fluctuations? More precisely, consider the (longitudinal) structure function of velocity difference across a length-scale $r$ :

$$S_p(r) \equiv \langle [\delta u(r)]^p \rangle, \tag{1a}$$

$$\text{where} \quad \delta u(r) \equiv [u_\alpha(\boldsymbol{x}+\boldsymbol{r}) - u_\alpha(\boldsymbol{x})] \frac{r_\alpha}{|\boldsymbol{r}|}. \tag{1b}$$

Here, $\boldsymbol{u}(\boldsymbol{x})$ is the velocity field as a function of the coordinates $\boldsymbol{x}$ and the symbol $\langle \cdot \rangle$ denotes averaging over the statistically stationary state of turbulence. Here and henceforth, we use the Einstein summation convention, repeated indices are summed. The $p$-th order structure function $S_p$ is the $p$-th moment of the probability distribution function (PDF) of velocity differences—if we know $S_p$ for all $p$ then we know the PDF. Typically, energy is injected into a turbulent flow at a large length scale $L$, while viscous effects are important at small length scales $\eta$, called the Kolmogorov scale, and dissipate away energy from the flow. In the intermediate range of scales $S_p(r) \sim r^{\zeta_p}$ where scaling exponents $\zeta_p$ are universal, i.e., they do not depend on how turbulence is generated. The dimensional arguments of Kolmogorov give $\zeta_p = p/3$, which also implies that the shell–integrated energy spectrum (distribution of kinetic energy across wavenumbers) $E(k) \sim k^{-5/3}$, where $k$ is the wavenumber. Experiments and direct numerical simulations (DNS) over last seventy years have now firmly established that the $\zeta_p(p)$ is a nonlinear convex function—a phenomenon called multiscaling or *intermittency*. Even within the Kolmogorov theory, turbulence is non-Gaussian because the odd-order structure functions (odd moments of

[1]Complex Fluids and Flows Unit, Okinawa Institute of Science and Technology Graduate University, Okinawa, Japan. [2]TIFR Centre for Interdisciplinary Sciences, Tata Institute of Fundamental Research, Gopanpally, Hyderabad, India. [3]Nordita, KTH Royal Institute of Technology and Stockholm University, Hannes Alfvéns väg 12, Stockholm, Sweden. ✉e-mail: marco.rosti@oist.jp

the PDF of velocity differences) are not zero. Intermittency is not merely non-Gaussianity, it implies that not only a few small order moments but moments of all orders are important in determining the nature of the PDF. We often write $\zeta_p = p/3 + \delta_p$, where $\delta_p$ are corrections due to intermittency. A systematic theory that allows us to calculate $\zeta_p$ starting from the Navier–Stokes equation is the goal of turbulence research.

Turbulent flows, both in nature and industry, are often multiphase, i.e., they are laden with particles, may comprise fluid mixtures, or contain additives such as polymers. Of these, polymeric flows are probably the most curious and intriguing: the addition of high molecular weight (about $10^7$) polymers in 10–100 parts per million (ppm) concentration to a turbulent pipe flow reduces the friction factor (or the drag) up to 5–6 times (depending on concentration)[1–3]. Evidently, this phenomenon, called turbulent drag reduction (TDR), cannot be studied in homogeneous and isotropic turbulent flows; nevertheless, polymer-laden homogeneous and isotropic turbulent (PHIT) flows have been extensively studied theoretically[4–6], numerically[7–23], and experimentally[24–28], to understand how the presence of polymers modifies turbulence, following the pioneering work by Lumley[29] and Tabor and de Gennes[30]. The simplest way to capture the dynamics of polymers in flows is to model the polymers as two beads connected by an overdamped spring with a characteristic time scale $\tau_p$. A straightforward parameterization of the importance of elastic effects is the Deborah number $De \equiv \tau_p/\tau_f$, where $\tau_f$ is some typical time scale of the flow. In turbulent flows, such a definition becomes ambiguous because turbulent flows do not have a unique time scale, rather we can associate an infinite number of time scales even with a single length scale[31–33]. In such cases, a typical timescale used to define De is the large eddy turnover time of the flow, $\tau_L$[23]. The phenomena of PHIT appear at high Reynolds and high Deborah numbers.

Research in polymeric flows turned in a novel direction when it was realized that even otherwise, laminar flows may become unstable due to the instabilities driven by the elasticity of polymers[34,35]. Even more dramatic is the phenomena of *elastic turbulence* (ET)[36], where polymeric flows at low Reynolds but high Deborah numbers are chaotic and mixing, with a shell–integrated kinetic energy spectrum $E(k) \sim k^{-\xi}$. It is still unclear whether this exponent is universal or not – experiments and DNS in two dimensions have obtained $3 \leq \xi \leq 4$, and theory[37] sets a lower bound with $\xi > 3$. A three-dimensional DNS of decaying homogeneous, isotropic turbulence with polymers additives (modeled as discrete dumbbells) also revealed an exponent $\xi \approx 4$ at late times (with a mild De dependence), when turbulence had sufficiently decayed and elastic stresses were dominant, likely marking the onset of ET[18]. In summary, as shown in Fig. 1, HIT (in Newtonian turbulence) appears at large Re and zero (and small) De; PHIT appears at large Re and intermediate De number, while ET appears at small Reynolds and large Deborah numbers.

Recently, experiments[24] and DNS[23] revealed an intriguing aspect of PHIT: The energy spectrum showed not one but two scaling ranges, a Kolmogorov-like inertial range at moderate wave numbers and a second scaling range with $E(k) \sim k^{-2.3}$ resulting purely due to the elasticity of polymers[23]. This is illustrated in the gray shaded region in Fig. 1. Even more surprising is the observation that both of these ranges have intermittency correction $\delta_p$ which are the same. This hints that even at low Re, where elastic turbulence (ET) appears, intermittent behavior may exist. In this paper, based on large resolution DNS of polymeric flows at low Reynolds number, we show that this is indeed the case.

## Results

We generate a statistically stationary, homogeneous, isotropic flow of a dilute polymer solution by the DNS of the Navier-Stokes equations coupled to the evolution of polymers described by the Oldroyd-B model:

$$\rho_f\left(\partial_t u_\alpha + u_\beta \partial_\beta u_\alpha\right) = -\partial_\alpha p + \partial_\beta\left(2\mu_f S_{\alpha\beta} + \frac{\mu_p}{\tau_p} C_{\alpha\beta}\right) + \rho_f F_\alpha, \quad (2a)$$

$$\partial_t C_{\alpha\beta} + u_\gamma \partial_\gamma C_{\alpha\beta} = C_{\alpha\gamma}\partial_\gamma u_\beta + C_{\gamma\beta}\partial_\alpha u_\gamma - \frac{1}{\tau_p}\left(C_{\alpha\beta} - \delta_{\alpha\beta}\right). \quad (2b)$$

Here, $\boldsymbol{u}$ is the incompressible solvent velocity field, i.e., $\partial_\beta u_\beta = 0$, $p$ is the pressure, $\mathcal{S}$ is the rate-of-strain tensor with components $S_{\alpha\beta} \equiv (\partial_\alpha u_\beta + \partial_\beta u_\alpha)/2$, $\mu_f$ and $\mu_p$ are the fluid and polymer viscosities, $\rho_f$ is the density of the solvent fluid, $\tau_p$ is the polymer relaxation time, and $\mathcal{C}$ is the polymer conformation tensor whose trace $C_{\gamma\gamma}$ is the total end-to-end squared length of the polymer. To maintain a stationary state, we inject energy into the flow using an Arnold-Beltrami-Childress (ABC) forcing, i.e., $\boldsymbol{F} = (\mu_f/\rho_f)[(A\sin z + C\cos y)\hat{\mathbf{x}} + (B\sin x + A\cos z)\hat{\mathbf{y}} + (C\sin y + B\cos x)\hat{\mathbf{z}}]$. The injected energy is ultimately dissipated away by both the Newtonian solvent ($\varepsilon_f$) and polymers ($\varepsilon_p$). The total energy dissipation rate, $\langle \varepsilon_T \rangle$, is given by:

$$\langle \varepsilon_T \rangle \equiv \langle \varepsilon_f \rangle + \langle \varepsilon_p \rangle \quad (3a)$$

$$\text{where} \quad \varepsilon_f \equiv \frac{2\mu_f}{\rho_f}\left(S_{\alpha\beta}S_{\alpha\beta}\right); \varepsilon_p \equiv \frac{\mu_p}{2\rho_f \tau_p^2}\left(C_{\gamma\gamma} - 3\right). \quad (3b)$$

We show typical snapshots of the two energy dissipation rates on two-dimensional slices of our three-dimensional DNS in Fig. 2. Details on numerical schemes and simulations are discussed in the Methods section.

The Newtonian ABC flow shows Lagrangian chaos in the sense that the trajectories of tracer particles advected by such a flow have sensitive dependence on initial condition[38]. Hence we expect that a polymer advected by the flow will go through a coil-stretch transition for large enough $\tau_p$. The back reaction from such polymers may give rise to elastic turbulence. The energy spectra of the Newtonian flows (and those for our non-Newtonian flows with small $\tau_p$) do not show any power-law range, and drop-off rapidly in wavenumber $k$, see Fig. (S1a) in the Supplementary Material. Beyond a certain value of $\tau_p$, the flow becomes chaotic, and the resulting flows with $De \gtrsim 1$ are able to sustain elastic turbulence. Henceforth we focus only on the flows that show ET.

Note an important difference between ET and usual Newtonian HIT. In the latter, the Kolmogorov length (or time) scale is defined as the scale where the inertial and viscous effects balance each other. Although we continue to use the same definition – $Re_\lambda$ and $\tau_K$ are calculated from the Newtonian DNS – these scales lose their usual meaning because ET appears at small Re at scales where the inertial term is negligible. The $\eta$ we obtain is, as expected, quite close to the scale of energy injection. Therefore, we use the box-size $L$, which is also the scale of energy injection, as our characteristic length scale.

We present our results for three different Deborah number flows with $De = 1$, 3, and 9 and Taylor scale Reynolds number $Re_\lambda \approx 40$. Let us begin by looking at the (shell–integrated) fluid energy spectrum

$$E(k) \equiv \int d^3\boldsymbol{m} \langle \hat{\boldsymbol{u}}(\boldsymbol{m})\hat{\boldsymbol{u}}(-\boldsymbol{m})\rangle \delta(|\boldsymbol{m}| - k), \quad (4)$$

where $\hat{\boldsymbol{u}}(\boldsymbol{m})$ is the Fourier transform of the velocity field $\boldsymbol{u}(\boldsymbol{x})$. We show the spectra for the three De numbers in Fig. 3a. The spectra $E(k)$ show power-law scaling over almost two decades when plotted on a log-log scale. Clearly, $E(k) \sim k^{-\xi}$ with $\xi = 4$ independent of the Deborah number. Note that in DNS of decaying PHIT, $\xi$ goes from 2.3 to 4 (and beyond as turbulence decayed) as time progresses[18]. While $\xi = 2.3$ was recently confirmed for PHIT via both DNS[23] and experiments[24], we now show via

DNS that ET is, in fact, a stationary state marked by $\xi = 4$ which is sustained by purely elastic effects, for a large enough polymer elasticity.

We have verified, using representative DNSs, that the scaling exponents of ET remain the same if we use the FENE-P model for polymers or a different forcing scheme[39], that is not white−in−time. We have also checked that reducing the resolution to $N^3 = 512^3$ reproduces the same spectra. Finally, we have turned off the advective nonlinearity in (2a) and also obtained the same spectra, thereby confirming that the turbulence we obtain is purely due to elastic effects. We plot all these spectra in Fig. (S1) of the Supplementary Material.

We also define the energy spectrum associated with polymer degrees of freedom as

$$E_p(k) \equiv \left(\frac{\mu_p}{\rho_f \tau_p}\right) \int d^3\boldsymbol{m} \left\langle \hat{B}_{\gamma\beta}(\boldsymbol{m}) \hat{B}_{\beta\gamma}(-\boldsymbol{m}) \right\rangle \delta(|\boldsymbol{m}| - k), \quad (5)$$

where the matrix $\mathcal{B}$ with components $B_{\alpha\gamma}$ is the (unique) positive symmetric square root of the matrix $\mathcal{C}$, defined by $C_{\alpha\beta} = B_{\alpha\gamma} B_{\gamma\beta}$[40,41]. We obtain $E_p(k) \sim k^{-\chi}$ with $\chi = 3/2$, as shown in the log-log plot of $E_p(k)$ in Fig. 3b. Note that the scaling range of $E_p(k)$ is somewhat smaller than that of $E(k)$. In the statistically stationary state of ET, the effect of the advective nonlinearity must be subdominant. Hence, at scales smaller

than the scale of the external force, the viscous term in the momentum equation must balance the elastic contribution[37]. Using a straightforward scaling argument, described in detail in the Supplementary Material, section IB, we obtain :

$$\xi = 2\chi + 1, \quad (6)$$

which is satisfied by the values of $\xi$ and $\chi$ we obtain. For further confirmation, we calculate all the contributions to the scale-by-scale kinetic energy budget in Fourier space (see the Supplementary Material, section IA). As expected, the contribution from the advective term in (2a) is negligible. Earlier theoretical arguments[37] have suggested $\xi > 3$, which has also been observed in experiments[42–44] – $\xi \approx 3.5$ over less than a decade of scaling range. We obtain $\xi \approx 4$, which satisfies the inequality and agrees with shell-model simulations[45]. Earlier theoretical arguments[37,46] had also assumed the same balance in the momentum equation that we have, but in addition, had assumed scale separation and a large-scale alignment of polymers in analogy with magnetohydrodynamics, obtaining $\xi = \chi + 2$, which is not satisfied by our DNS.

Note further that experiments often obtain power-spectrum as a function of frequency, and they can be compared with power-spectrum as a function of wavenumber (typically obtained by DNS) by using the Taylor "frozen-flow" hypothesis[47]. In the absence of a mean flow and negligible contribution from the advective term it is not a priori obvious that the Taylor hypothesis should apply to ET. We have confirmed from our DNS that a frequency-dependent power spectrum obtained from a time series of velocity at a single Eulerian point also gives $\xi = 4$ (see the Supplementary Material, Fig. (S1b)).

### Second order structure function

Next, we consider the second-order structure function, $S_2(r)$, which is the inverse Fourier transform of $E(k)$. This requires some care. As a background, let us first consider the case of HIT (Newtonian homogeneous and isotropic turbulence). Let us ignore intermittency and concentrate on scaling a-la Kolmogorov. The second-order structure function is expected to have the following form

$$S_2(r) \sim \begin{cases} r^{\zeta_2} & \text{for} \quad L > r > \eta, \\ r^2 & \text{for} \quad \eta > r > 0. \end{cases} \quad (7)$$

The range of scales $L > r > \eta$ is the inertial range. Let us remind the reader that the behavior $S_2 \sim r^2$ for small enough $r$ follows from the assumption that the velocities are analytic functions of coordinates, which must always hold for any finite viscosity, however small. We call $S_2 \sim r^2$ the *trivial* scaling. The strategy to extract the exponent $\zeta_2$ from DNS is to run simulations at higher and higher Reynolds

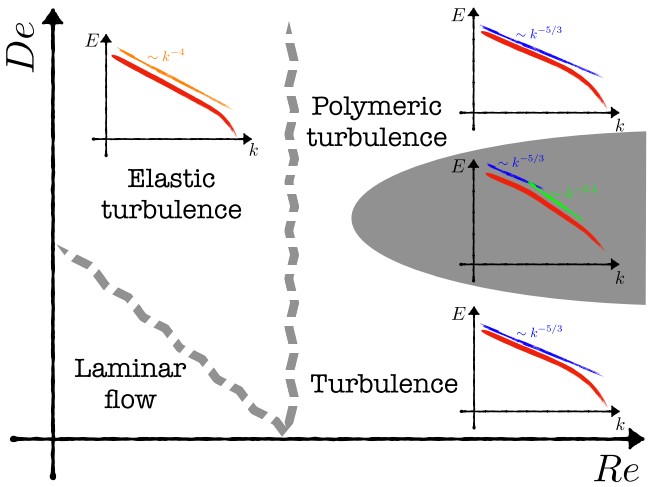

**Fig. 1 | Polymeric flows.** An illustrative sketch of the different regimes of polymer-laden flows: classical Newtonian *turbulence* (HIT) at large Re and zero (or small) De, *polymeric turbulence* (PHIT) at large Re and intermediate De, and *elastic turbulence* (ET) at small Re and large De. The shaded region shows the recently observed elastic scaling regime in addition to the classical Kolmogorov scaling[23,24].

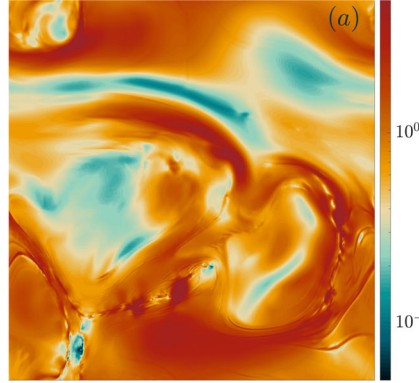
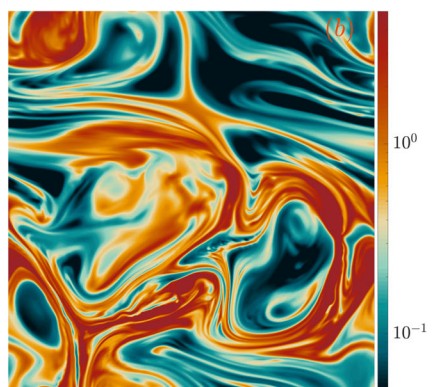

**Fig. 2 | Flow visualizations.** Two-dimensional slices of the three-dimensional domain showing snapshots of the normalized (**a**) fluid dissipation field $\varepsilon_f / \langle \varepsilon_f \rangle$ and of (**b**) the polymer dissipation field $\varepsilon_p / \langle \varepsilon_p \rangle$ in ET for De = 9.

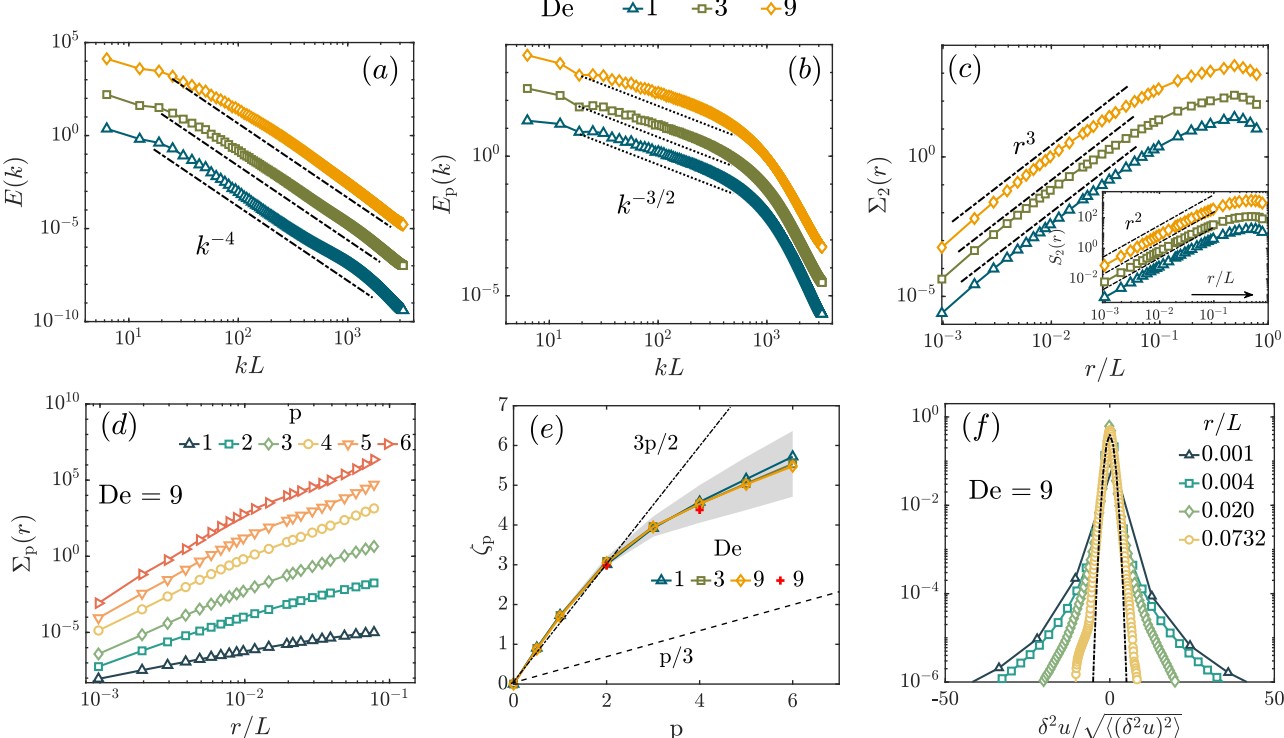

**Fig. 3 | Spectra and structure functions. a** The fluid energy spectra show a universal scaling $E(k) \sim k^{-4}$ independent of De. A steeper than $k^{-3}$ fall-off of the spectrum means the velocity fields are smooth; $S_2(r) \sim r^2$ for small $r$, shown in the inset of panel (c). **b** The polymer spectra $E_p(k) \sim k^{-3/2}$ follows from the scaling of $E(k)$. **c** Plot of the second order structure function of second differences, which scale as $\Sigma_2(r) \sim r^3$. The exponent is the same for the different De, although the range of scaling depends weakly on De. The inset shows the analytic scaling of $S_2(r)$. **d** Structure-function of second differences, $\Sigma_p$, for various orders $p$ for De = 9. **e** The exponents $\zeta_p$ versus $p$, calculated from the scaling behavior of $\Sigma_p$. Departure from the straight line $\zeta_p = 3p/$ 2 shows intermittency. The shaded region shows the standard deviation on the exponents computed from 18 snapshots. The two red + symbols mark the exponents $\zeta_2$ and $\zeta_4$ obtained with an alternate forcing scheme[39]. That they lie well within error bars goes on to show that the results are independent of the large scale forcing. **f** Probability distribution function of $\delta^2 u(r)$ for four different values of $r$ at De = 9. The distributions are non-Gaussian at small separations, while they become closer to a Gaussian (shown as a black dash-dotted curve) for large $r$. The corresponding cumulative distribution functions, computed using the rank-order method, are shown in the Supplementary Material, section IIIB.

numbers, which means smaller and smaller $\eta$ to obtain a significant inertial range from which $\zeta_2$ can be extracted. In Kolmogorov theory $\zeta_2 = 2/3$, we call this the *non-trivial* scaling. The theory of ET is much less developed than that of HIT. Nevertheless, we may assume that the velocities must still be analytic functions. Hence for ET the following must hold

$$S_2(r) \sim r^2 \quad \text{for} \quad r \to 0. \tag{8}$$

As this scaling follows directly as a consequence of the analyticity of velocities, we again call this trivial scaling. If there is a non-trivial scaling of second order structure function in ET – here we are not talking about intermittency corrections to a non-trivial scaling but just the existence of the non-trivial scaling exponent – it may show up in the following manner

$$S_2(r) \sim \begin{cases} r^{\zeta_2} & \text{for} \quad L > r > \ell, \\ r^2 & \text{for} \quad \ell > r > 0. \end{cases} \tag{9}$$

This requires the introduction of a new length scale $\ell$ which cannot depend on Reynolds number because in ET we are already in the range of small and fixed Reynolds number. The scale $\ell$ may depend on the Deborah number. To check if it does, we plot $S_2(r)/r^2$ for three different De ranging from 1 to 9 in the Supplementary Material Fig. (S3b). At small enough $r$ they all show $S_2 \sim r^2$. As $r$ increases they all depart from this trivial scaling at a length scale $\ell$ which depends very weakly on De, if at all. This implies that even if a non-trivial scaling for $S_2$ exists in ET, it may require DNS at impossibly high De to be able to extract $\zeta_2$. Nevertheless, we have now demonstrated that in ET:

(a) $S_2$ shows trivial scaling at small $r$, i.e., the velocity field is analytic, and

(b) there is a departure from the trivial scaling.

Does the departure from the trivial scaling show a new scaling range? To explore this possibility we plot on a log-log scale $S_2(r)$ as a function of $r$ in the Fig. (S3a) of the Supplementary Material, but it is unclear if there is a clear scaling range at intermediate $r$. Even if there is a non-trivial scaling exponent, it cannot be detected from the data, which is the highest resolution DNS of ET done so far.

It often helps to detect a scaling range if we know beforehand what the scaling exponent is. In ET, unlike HIT, there is no theory that tells us what $\zeta_2$ should be, but we do know that the Fourier spectrum of energy behaves like $E(k) \sim k^{-\xi}$, with $\xi \approx 4$ (see Fig. 3a). Usual straightforward power counting implies that $S_2(r) \sim r^{\xi-1} \sim r^3$ (see section IB of Supplementary Material for details), while we obtain $S_2 \sim r^2$. This paradox is resolved by noting that, in the limit $r \to 0$, $r^3$ is subdominant to $r^2$, hence $S_2 \sim r^2$ as $r \to 0$ for any velocity field whose spectra $E(k) \sim k^{-\xi}$ with $\xi > 3$, see, e.g., Ref. [48, Appendix G]. This is also known from the direct cascade regime of two-dimensional turbulence (with Ekman friction) where $E(k) \sim k^{-\gamma}$ with $\gamma > 3$ and $S_2(r) \sim r^2$, see, e.g., Refs. [49,50, page 432]. This suggests that $S_2(r)$ satisfies (9) with $\zeta_2 = \xi - 1 \approx 3$. To test this, we plot the compensated second-order structure function $S_2(r)/r^3$ as a function of $r$ in the Supplementary Material Fig. (S3c). We detect no range at small or intermediate $r$ where this non-trivial scaling holds.

Now we consider the possibility that $S_2(r) \sim A r^2 + B r^3 +$ h. o. t. Here, the symbol h. o. t denotes higher order terms in $r$. We use a trick[51,52] to extract the subleading term, which scales with the non-trivial scaling exponent: the idea is to remove the analytic contribution by considering the *second difference* of velocities:

$$\delta^2 u(r) \equiv [u_\alpha(\mathbf{x}+\mathbf{r}) - 2u_\alpha(\mathbf{x}) + u_\alpha(\mathbf{x}-\mathbf{r})]\left(\frac{r_\alpha}{r}\right), \qquad (10a)$$

$$\text{and define} \quad \Sigma_2(r) \equiv \left\langle (\delta^2 u)^2 \right\rangle. \qquad (10b)$$

We plot $\Sigma_2(r)$ in Fig. 3c. We find that $\Sigma_2(r)$ shows a significant scaling range as $r \to 0$ with the non-trivial scaling exponent $\zeta_2 \approx 3$. This implies that, in ET the velocity fluctuations across a length scale $r$ can be expanded in an asymptotic series in $r$ as

$$\langle \delta u_\alpha(\mathbf{r}) \rangle \equiv \langle u_\alpha(\mathbf{x}+\mathbf{r}) - u_\alpha(\mathbf{x}) \rangle \sim G_{\alpha\beta} r_\beta + H_{\alpha\beta} r_\beta^h + \text{h.o.t.}, \qquad (11)$$

where $G_{\alpha\beta}$ and $H_{\alpha\beta}$ are (undetermined) expansion coefficients, and $h \approx \zeta_2/2 = 3/2$. The use of $\Sigma_2$ is necessary to extract the subleading contribution.

To appreciate the importance of this result, let us revisit the Kolmogorov theory of turbulence: in the limit $r \to 0$ at a finite viscosity $\mu_f$, $\langle \delta u_\alpha(r) \rangle \sim r$ since velocity gradients are finite. But if we first take the limit $\nu \to 0$ and then $r \to 0$ ($\nu \equiv \mu_f/\rho_f$ is the kinematic viscosity) $\langle \delta u_\alpha(r) \rangle \sim r^h$ with $h \approx 1/3$. The velocity field is *rough*. In contrast, ET is, by definition, a phenomenon at a finite viscosity (small Reynolds number), thus, the limit $\nu \to 0$ does not make sense – the velocity field is always smooth. But the non-trivial nature of ET manifests itself in the first subleading term in the expansion (11), and this is best revealed not by the velocity differences, but by the second difference of velocity. This is the first important result of our work.

### Intermittency based on velocity differences

The crucial lesson to learn from the previous section is that in ET, to uncover the non–trivial scaling of velocity differences we must use the second differences of velocity rather than the usual first difference. Other than this peculiarity, the rest of this section follows the standard techniques[47] used to study intermittency/multifractality.

We define the $p$-th order structure function of the second difference of velocity across a length scale $r$ as:

$$\Sigma_p(r) \equiv \left\langle |\delta^2 u(\mathbf{r})|^p \right\rangle. \qquad (12)$$

We show a representative plot of $\Sigma_p$'s for all integer $p = 1, \ldots, 6$ in Fig. 3d for De = 9. Clearly, there exists a scaling regime for which the scaling exponents $\zeta_p$ can be extracted by fitting $\Sigma_p(r) \sim r^{\zeta_p}$ as $r \to 0$. The scaling exponents as a function of $p$ are shown in Fig. 3e, where we have also included half-integer values of $p$.

To obtain reasonable error bars on $\zeta_p$, we have proceeded in the following manner: first, we find a suitable scaling regime for each order by visual inspection; next, in these chosen ranges, we find the local slopes of the log-log plot of $\Sigma_p(r)$ vs $r$, to obtain $\zeta_p$ as a function of $r$: $\zeta_p(r) = (\Delta \log \Sigma_p(r))/(\Delta \log r)$. This process is repeated for multiple time snapshots (two successive snapshots are separated by at least one eddy turnover time) of the velocity field data. The mean value over the set of exponents thus obtained is the exponent $\zeta_p$ in Fig. 3e and the standard deviation sets the error bar which is shown as a shaded region. Clearly, $\zeta_p$ is a non-linear function of $p$. This unambiguously establishes the existence of intermittency in ET.

Furthermore, we have confirmed that the structure functions $S_p$ of even order up to 6 grow as $r^p$ for small $r$, see the Supplementary Material section IIC for discussion. Thereby we confirm, following prescription in Ref. 53, that the structure functions of all orders are

analytic. The structure functions begin to depart from this analytic scaling at a scale that depends very weakly on De (if at all), but this scale decreases as $p$ increases. We also use another forcing scheme[39] and calculate the exponents $\zeta_p$ for $p = 2$ and 4, marked as two + symbols in red color in Fig. 3e. Within errorbars they agree with the values we have obtained suggesting that the $\zeta_p$ are universal.

Let us again emphasize that intermittency is a fundamental property of structure functions, both $S_p$ and $\Sigma_p$. The use of $\Sigma_p$ is merely to help us extract the exponents $\zeta_p$.

**PDF of velocity differences.** Another way to demonstrate the effects of intermittency is by looking at the PDF of velocity differences across a length scale. From the structure function we have obtained intermittent behavior for scales $r/L < 1$. Thus, we expect the PDF of velocity differences to be close to Gaussian for $r/L \approx 1$ and to have long tails (decaying slower than Gaussian) for $r/L < 1$. This indeed is the case, as is shown in Fig. 3f where we plot the PDFs of $\delta^2 u(r)$ for different separations $r$ (for De = 9). The tails of the distribution of $\delta^2 u$ decay much slower than Gaussian, thereby clearly demonstrating intermittency.

Note that also the PDF of the usual velocity differences $\delta u(r)$ is non-Gaussian, see the Supplementary Material Fig. (S8). But the PDF of $\delta^2 u$ falls off slower than $\delta u$ at the same $r$, in other words larger fluctuations are more likely to appear in the second difference of velocity – it is more intermittent. This non-Gaussianity of probability distributions can be quantified by the Kurtosis (also called Flatness) defined by

$$\mathcal{K}(r) = \frac{\left\langle [\delta u(r)]^4 \right\rangle}{\left\langle [\delta u(r)]^2 \right\rangle^2} \quad \text{and} \quad \mathcal{K}_2(r) = \frac{\left\langle \left[\delta^2 u(r)\right]^4 \right\rangle}{\left\langle \left[\delta^2 u(r)\right]^2 \right\rangle^2}, \qquad (13)$$

for the first and second difference of velocity, respectively. For Gaussian distributions, the Kurtosis is 3. We find $\mathcal{K}_2 \approx 3$ as $r \to L$, i.e., the PDFs (of $\delta^2 u$) are close to Gaussian for large separations. From the scaling behavior of structure function we obtain $\mathcal{K}_2(r) \sim r^{\zeta_4 - 2\zeta_2}$ as $r \to 0$, which is consistent with $\mathcal{K}_2(r) \sim r^{-1.6}$ obtained from the distribution of second differences shown in Fig. 4a. Furthermore, we find that the Kurtosis is independent of De. The gray shaded region marks the range used to compute the scaling exponents for $\mathcal{K}_2(r)$. We obtain: $-1.6 \pm 0.3$, $-1.6 \pm 0.1$, and $-1.6 \pm 0.1$ for De = 1, 3 and 9 respectively. This is further evidence in support of universality of intermittency in ET. The Kurtosis of first difference $\mathcal{K}(r)$ is also close to a Gaussian as $r \to L$, but grows much slower than $\mathcal{K}_2(r)$ as $r \to 0$ as shown in Fig. 4b. In Newtonian HIT at high Re, the most recent DNS[54,55] shows $\zeta_4 - 2\zeta_2 \approx -0.12$ so that $\mathcal{K} \sim r^{-0.12}$. Hence, the intermittency we obtain in ET is more intense than what is observed in HIT.

We also calculate the cumulative PDF (CDF) of $\delta^2 u$ by rank–order method, thereby avoiding the usual binning errors that appear while calculating PDFs via histograms. In section IIIB of the Supplementary Material, we show that rescaling the abscissa of the CDFs by the root-mean-square value of $\delta^2 u$ does not collapse the CDFs for different $r$, i.e., the PDFs are not Gaussian.

Altogether the PDFs of $\delta^2 u$ provide us with three additional evidences in support of intermittency in ET.

### Intermittency based on dissipation

In HIT (high Re, Newtonian turbulence) there are two routes to study intermittency: one is through structure functions and another is through the fluctuations of the energy dissipation rate[47] – the PDF of $\varepsilon_f$ deviates strongly from a log-normal behavior[56]. We now take the second route for ET, in which case there are two contributions to the total energy dissipation – $\varepsilon_f$ and $\varepsilon_p$. In Fig. 5a and b we plot the PDFs of the logarithm of $\varepsilon_f$ and $\varepsilon_p$, respectively. We find that the former decays slower than a Gaussian, i.e., the PDF itself falls off slower than a log-

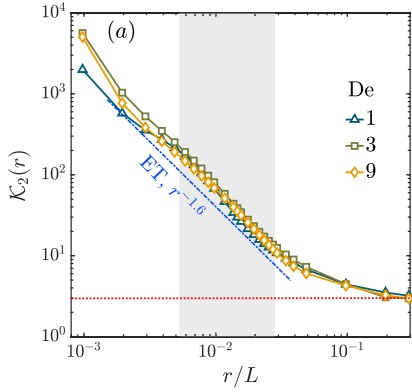
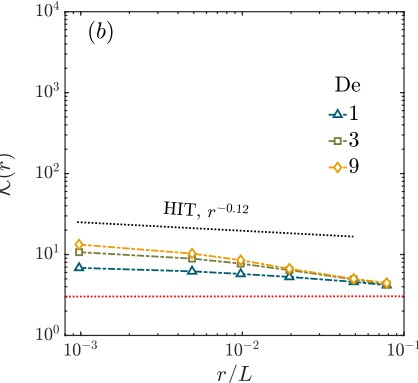

**Fig. 4 | Kurtosis.** The kurtoses, (**a**) $\mathcal{K}_2$ and (**b**) $\mathcal{K}$ as a function of the scale $r$ for De = 1, 3 and 9. The red dashed line is at ordinate equal to 3. We also show in (**a**) a line of slope −1.6. The scaling exponent of kurtosis, obtained from fitting the data in the gray-shaded region, are: −1.6 ± 0.3, −1.6 ± 0.1, and −1.6 ± 0.1, for De = 1, 3, and 9, respectively. This demonstrates both the non-Gaussian nature of the PDFs and the universality of the exponents with respect to De. The kurtosis of $\delta u$, $\mathcal{K}$, grows slower as $r \to 0$ and may not be universal. To compare, we also plot, in (**b**), the corresponding result for Newtonian HIT. Both the kurtoses $\mathcal{K}, \mathcal{K}_2 \to 3$ (shown in dotted-red line) as $r \to L$. This indicates that at large separations the statistics of velocity difference are close to a Gaussian.

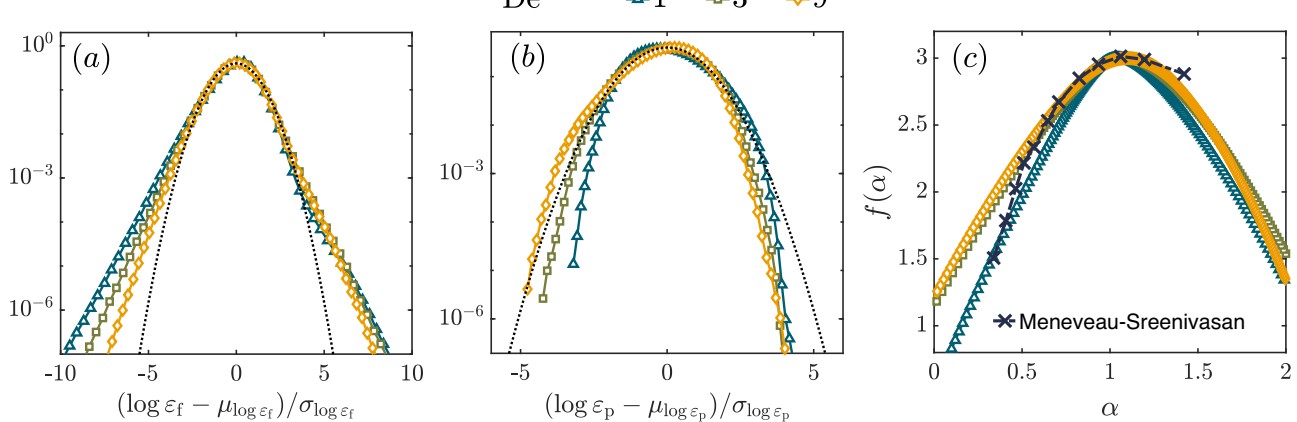

**Fig. 5 | Dissipation rates. a** PDFs of the logarithm of the fluid energy dissipation rate $\varepsilon_\mathrm{f}$ for all three De numbers. We denote by $\mu_{\log \varepsilon_{\mathrm{f/p}}}$ and $\sigma_{\log \varepsilon_{\mathrm{f/p}}}$ the mean and variance of the logarithm of energy dissipation rates. The distributions deviate significantly from a log-normal behavior in both the left and right tails. The right tails coincide for large De, similar to the coincident right tails at large Re in Newtonian HIT. **b** PDFs of the logarithm of the polymer energy dissipation rate $\varepsilon_\mathrm{p}$ for all three De numbers. The PDFs of $\log \varepsilon_\mathrm{p}$ are sub-Gaussian, i.e., decay faster than a Gaussian, indicating $\varepsilon_\mathrm{p}$ is not intermittent. **c** The multifractal spectra of the fluid dissipation field calculated from the scaling of the energy dissipation rate $\varepsilon_r$ calculated over a cube of side $r$. The black dash-dotted line shows the spectrum for Newtonian HIT[57].

normal, whereas the latter decays faster than a Gaussian. The fact that the PDF of $\varepsilon_\mathrm{p}$ falls off much faster than that of $\varepsilon_\mathrm{f}$ can even been seen by comparing Fig. 2a with Fig. 2b. Clearly, the statistics of $\varepsilon_\mathrm{p}$ are non-intermittent. Henceforth, following the standard analysis pioneered by[57] for HIT, we study the scaling of the $q$-th moment of the viscous dissipation averaged over a cube of side $r$,

$$\langle \varepsilon_r^q \rangle \sim r^{\lambda_\mathrm{q}}, \quad \text{where} \tag{14a}$$

$$\varepsilon_r \equiv \frac{2\mu_\mathrm{f}}{\rho_\mathrm{f}} \left\langle S_{\alpha\beta} S_{\alpha\beta} \right\rangle_r. \tag{14b}$$

Here the symbol $\langle \cdot \rangle_r$ denotes averaging over a cube of side $r$. The Legendre transform of the function $\lambda_\mathrm{q}$ gives the multifractal spectrum (also called the Cramer's function) $f(\alpha)$:

$$\lambda_\mathrm{q} = \inf_\alpha [q(\alpha - 1) + 3 - f(\alpha)], \tag{15}$$

where singularities in the dissipation field with exponent $\alpha - 1$ lie on sets of dimension $f(\alpha)$. We plot the $f(\alpha)$ spectrum for ET in Fig. 5c.

There are minor differences between the multifractal spectrum for De = 3 and 9 on one hand and De = 1 on the other hand. The clear collapse of the multifractal spectra at large De hints towards a universal multifractality in ET in the limit of large De. For comparison, we also plot, in Fig. 5c the multifractal spectrum for HIT as a black dash-dotted curve[57]. In HIT the intermittency model based on velocity is closely connected to the intermittency models based on dissipation[47]. The development of such a formalism for ET, although important, is not considered in this work.

## Discussion

We note that the phenomenon of elastic turbulence has no Newtonian counterpart – in the absence of the polymers, this phenomenon disappears. Nevertheless, as HIT is the model of turbulence that has been studied in great detail we have used it as an illustrative example to compare with ET. Such comparison must be done with care. In HIT, the theory of Kolmogorov helps us understand the simple scaling of the energy spectrum, although a systematic derivation starting from the Navier–Stokes equation is still lacking. The key insight of Kolmogorov's theory is that the energy flux across scales, due to the

nonlinear advective term, is a constant. In practice, the flux is a fluctuating quantity, where its mean value determines the simple scaling prediction $\zeta_p = p/3$, while the fluctuations of the flux is the reason behind intermittency. The fluctuations of the flux show up as fluctuations of the energy dissipation rate (because the advective term conserves energy), which is multifractal.

Elastic turbulence was first discovered at the start of this century. Almost all studies of ET, so far, have concentrated on understanding the scaling of the energy spectrum. A theory at the level of Kolmogorov's theory for HIT is still lacking. Nevertheless, it is clear that the mechanism of ET is very different from HIT. In the latter, it is the nonlinear advective term that is responsible for turbulence, while in the former the advective term is expected to be subdominant, it is the stress from the polymers that must balance the viscous dissipation in the range of scales where ET is found. We show that this is indeed the case. A consequence of this balance is that the scaling exponents of $E(k)$ and $E_p(k)$ are related to each other by (6). In ET there is not one but two possible mechanisms of energy dissipation. A particularly intriguing result we obtain is that only one of them, $\varepsilon_f$, shows intermittent behavior since the energy dissipation rate due to the polymers is not intermittent, and its logarithm remains sub-Gaussian.

In summary, we have shown that both the velocity field and the energy dissipation field in ET are intermittent/multifractal. But this multifractality is very different from the multifractality seen in HIT. In HIT, in the limit of viscosity going to zero, the velocity field is rough. In contrast, the velocity field in ET is smooth at leading order, and roughness and multifractal behavior appear due to the sub-leading term. Consequently, although the velocity difference across a length scale is intermittent, it is necessary to use the second difference of velocity to properly reveal the intermittency. Finally, note that in HIT, the multifractal exponents are expected to be universal, i.e., they are independent of the method of stirring and the Reynolds number (in the limit of large Reynolds number). In ET the multifractality appears at small Re and large De > 1. All the evidence from our DNSs suggest that intermittency in ET is also universal with respect to Deborah number, method of stirring, and choice of model of polymers, although significant future work with high-resolution DNSs is necessary to provide conclusive evidence.

## Methods

We solve eqns. (2a), (2b) using a second order central-difference scheme on a $L = 2\pi$ tri-periodic box discretized by $N^3 = 1024^3$ collocation points, such that $L/N = \Delta \approx 0.05\eta$, where $\eta \equiv (\nu^3/\langle \varepsilon_f \rangle)^{1/4}$ is the Kolmogorov dissipation length scale and $\nu \equiv \mu_f/\rho$ is the kinematic viscosity. Integration in time is performed using the second order Adams-Bashforth scheme with a time step $\Delta t \approx 10^{-5}\tau_K$, with $\tau_K \equiv \sqrt{\nu/\langle \varepsilon_f \rangle}$. We use 18 snapshots in our analysis, with successive snapshots separated by $\approx 1.6 \times 10^4 \tau_K$. We choose a $\mu_f$ so as to obtain a laminar flow in the Newtonian case with $A = B = C = 1$ – this corresponds to the Taylor scale Reynolds number $Re_\lambda \approx 40$. Next, we choose $\mu_p$ such that the viscosity ratio $\mu_f/(\mu_f + \mu_p) = 0.9$ – this corresponds to dilute polymer solutions[14], and we vary $\tau_p$ over two order of magnitudes. The numerical solver is implemented on the in-house code *Fujin*; see https://groups.oist.jp/cffu/code for additional details and validation tests. The very same code has been successfully used on various problems involving Newtonian and non-Newtonian fluids[58–61].

### Reporting summary

Further information on research design is available in the Nature Portfolio Reporting Summary linked to this article.

## Data availability

All data needed to evaluate the conclusions are present in the paper and/or the Supplementary Materials and available as a Source Data file. Source data are provided in this paper.

## Code availability

The code used for the present research is a standard direct numerical simulation solver for the Navier–Stokes equations. Full details of the code used for the numerical simulations are provided in the Methods section and references therein.

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

## Acknowledgements

The research was supported by the Okinawa Institute of Science and Technology Graduate University (OIST) with subsidy funding from the Cabinet Office, Government of Japan. The authors acknowledge the computer time provided by the Scientific Computing & Data Analysis section of the Core Facilities at OIST and the computational resources provided by the HPCI System (Project IDs: hp210229, hp210269, and hp220099). The authors RKS and MER thank Prof. Guido Boffetta for crucial insights and suggestions and for bringing to our notice Ref. 51. PP acknowledges support from the Department of Atomic Energy (DAE), India under Project Identification No. RTI 4007, and DST (India) Project No. MTR/2022/000867. DM acknowledges the support of the Swedish Research Council Grant No. 638-2013-9243. NORDITA is partially supported by NordForsk. DM gratefully acknowledges the hospitality of OIST.

## Author contributions

M.E.R. and D.M. conceived the original idea. M.E.R. planned and supervised the research, and developed the code. R.K.S. and M.E.R. performed the numerical simulations. R.K.S. analyzed the data. R.K.S. and D.M. wrote the first draft of the manuscript, with inputs from P.P. and M.E.R. All authors outlined the manuscript content and wrote the manuscript.

## Competing interests

The authors declare that they have no competing interests.
