## [Peer Review File · Nature Communications]

REVIEWER COMMENTS

Reviewer #1 (Remarks to the Author):

This paper provides insights into the intermittent behavior of elastic turbulence (ET) and compares it with the behavior of homogeneous isotropic Newtonian turbulence (HIT). The authors demonstrate that the velocity field and energy dissipation field in ET exhibit intermittent and multifractal behavior. However, this multifractality differs from the multifractality observed in HIT. In HIT, the velocity field is rough, whereas in ET, it is smooth at the leading order. The roughness and multifractal behavior in ET arise from sub-leading terms.

In summary, the paper is well-written, and the authors investigate the dynamics of elastic turbulence in a non-smooth flow and analyze the intermittency in the flow. However, there is a lack of detailed analysis of the physical reasons behind this intermittent behavior, for instance the role of the polymer stress in ET. The authors need to gain further insight into this problem. It should be noted that the similar problem has already been studied by Watanabe and Gotoh (PoF, 2014) with a better finite extension model of the polymers (FENE-P vs the Oldroyd-B model used in the current study). Thus, the manuscript does not meet the high standard of Nature Communications. If the authors want to publish the manuscript elsewhere, the following comments should be addressed:

1. How are the Taylor scale Reynolds number and Kolmogorov scale defined? Are they defined for each ET case or based on the Newtonian counterpart?
2. FIG. 2 is not mentioned in the maintext.
3. Watanabe and Gotoh (PoF, 2014) reported that the energy spectrum in decaying isotropic turbulence, with a comparable Reynolds number used in this study, obeyed the power law scaling $E(k) \sim k^{(-\alpha)}$. However, they found that α increases with the Weissenberg number. In this study, the scaling exponent α seems universal. This difference needs some further discussion.
4. From the manuscript, I am not convinced by the authors that it is necessary to use the second difference of the velocity to study the intermittency. In Fig. S3(b), the fourth and sixth order velocity structure functions already departure from the smooth scaling, i.e., $S_p \sim r^p$. The scaling exponents of S_p , the PDF and kurtosis of the velocity increment should be presented before the use of the second difference of the velocity.

5. This paper presents an unconventional definition of structure function that utilizes the second difference of velocity to extract the scaling exponents. However, the paper lacks comparisons or discussions with widely-known Newtonian baselines in the field. It would be valuable to compare the results with existing theories or experimental observations to provide a broader context for the findings.

6. Is there a critical scale for the polymers to affect elastic turbulence, or can it be inferred from the kinetic energy spectrum? Do the results in this paper imply that we should resolve scales much smaller than the Kolmogorov scale in ET?

7. The authors should provide more details about the simulations, such as how the time step of about $10^{-5} \tau_K$ was chosen and the number of snapshots for data analysis.

8. In section III. RESULTS, the authors argue that the scaling in the kinetic energy spectrum and the polymer spectrum follows $\xi=2\chi+1$. It would be helpful to explain the physical meaning behind this relationship.

9. It is difficult for me to understand the statement "... $\zeta_2=3$. This is consistent with $E(k)$ we obtained from our simulations." Can the authors provide further explanation?

10. In figure 3(d), it seems that there two scaling ranges for $r>\eta$ and $r<\eta$. There should be more discussions on this.

11. The definition of $G_{\alpha\beta}$ and $H_{\alpha\beta}$ in equation (9) requires further explanation.

12. In the supplemental material: II. Fluxes, the definitions of (S1a) and (S1b) appear to be opposite.

13. Most figures only show the results for $De = 1, 3, \text{ and } 9$. It would be beneficial to include HIT results for comparison. For example, in the supplemental material: III., the authors argue that the initial flat range disappears completely at $p = 6$ for $De = 9$. More data should be presented here to determine whether this phenomenon arises from elastic effects or high-order numerical errors.

14. The format of the references does not meet Nature Communication's requirements. For example: Physical Review Letters sometimes appears as "Physical review letters", some other times appears as "Phys. Rev. Lett.". Some of the authors' first names are abbreviated, some are not.

Reviewer #3 (Remarks to the Author):

I read the paper with interest and curiosity. It is well written and the results are novel and interesting.

In the following I have two comments which the authors should take into account for the paper to be published.

1) It is said in the paper that TDR (turbulent drag reduction) is observed in PHIT situation (large De and large Re). Actually,

for pipe flow TDR is observed somehow in the intermediate region between PHIT and HIT. It is worthwhile to recall that finite extension of polymers, usually taken into account in the FENE-P models, is crucial to observe the Maximum Drag Reduction

Asymptote (MDR) by increasing the polymer concentration. In Oldroyd model concentration plays no role and MDR cannot be observed. I suggest the authors to clarify this point to avoid confusion in the present literature. After all, their research has nothing to do with TDR.

2) From a more technical point of view, denoting by $\delta v(r)$ the difference $v(x+r)-v(x)$, then the quantity discussed by the author is $\delta v(r)+\delta v(-r)$. For the second order structure function we have:

$$\Sigma_2(r) \langle [\delta v(r)+\delta v(-r)]^2 \rangle = 2 S_2(r) + 2 \langle \delta v(r)\delta v(-r) \rangle$$

The quantity $\Sigma_2(r)$ is the superposition of regular scaling r^2 coming from the first contribution and something else.

The first term of the above expression goes as r^2 as shown by the author in the insert of figure (3c). So the intermittent scaling should be given by the second term (at least for small r).

It could be useful to check that this is the case.

In other words I suggest that the author explicitly show that $\langle \delta v(r) \delta v(-r) \rangle$ displays anomalous scaling.

This can be easily done by showing $\Sigma_2(r) \sim 2S_2(r)$. Eventually they should explain why this is not possible.

Referee Report on NCOMMS-23-42570-T “Intermittency in the not-so-smooth elastic turbulence”

This manuscript reports on intermittency in Elastic Turbulence (ET) in the low Reynolds number $Re \sim 1$, high Deborah number $De \gg 1$ limit. Since intermittency is regarded as a high Reynolds number phenomenon, this low- Re , ET-intermittency, although occurring sub-dominantly, appears surprising. The intermittency is quantified using spectra, probability density functions (PDF) of second differences of the velocity field and their moments, and the multi-fractal spectrum. The results seem to be cogently presented and the main result appears to be non-trivial. However, following are the issues I would like the authors to address.

1. The flow is energized using an Arnold-Beltrami-Childress (ABC) forcing. How do the small-scale sub-leading intermittent statistics depend on the specifics of the forcing scheme? Can the authors comment on the generality of the results obtained ?
2. There appears no discussion of the Flow visualization Fig. 2 in the paper. It would be useful to connect Fig. 2 with the PDF discussion of Fig. 4 (a),(b).
3. The Kurtosis of the distributions of velocity second differences given in the Supplementary Material (SM) is shown to scale as $r^{-1.5}$. This corresponds to a more precipitous drop with increasing scale-size than seen in classical turbulence which the authors quote as $r^{-0.11}$ (the authors must include a more recent flatness number for classical turbulence). The authors may want to comment on the difference between the two exponents and its implications.
4. In order for this interesting work to merit publication in Nature Communications the authors must surely expend greater effort into informing the reader about the importance and implications of this work.

Overall Comments

I believe that with suitable revisions along the suggested lines this manuscript can be considered for publication in Nature Communications.

Reviewer 1:

This paper provides insights into the intermittent behavior of elastic turbulence (ET) and compares it with the behavior of homogeneous isotropic Newtonian turbulence (HIT). The authors demonstrate that the velocity field and energy dissipation field in ET exhibit intermittent and multifractal behavior. However, this multifractality differs from the multifractality observed in HIT. In HIT, the velocity field is rough, whereas in ET, it is smooth at the leading order. The roughness and multifractal behavior in ET arise from sub-leading terms.

In summary, the paper is well-written, and the authors investigate the dynamics of elastic turbulence in a non-smooth flow and analyze the intermittency in the flow.

1. **Reviewer 1** However, there is a lack of detailed analysis of the physical reasons behind this intermittent behavior, for instance the role of the polymer stress in ET. The authors need to gain further insight into this problem.

Response: We do analyze the role of the polymer stress – we calculate its contribution to the energy flux and relate the scaling exponent of kinetic energy to that of polymer spectra. We also analyze the role of energy dissipation due to the presence of polymers.

A detailed analysis of the physical reasons behind the intermittent behaviour is possible only if a theory can be developed. As there is, so far, no theory of intermittency even in usual Newtonian HIT it is unfair to ask for such a theory at the outset. It is likely that such a theory for intermittency in ET will take decades to develop. We, for the first time, show that there exists intermittency in ET.

2. **Reviewer 1** It should be noted that the similar problem has already been studied by Watanabe and Gotoh (PoF, 2014) with a better finite extension model of the polymers (FENE-P vs the Oldroyd-B model used in the current study). Thus, the manuscript does not meet the high standard of Nature Communications.

Response: We have to disagree with the Reviewer on this point, since we consider our work very different. Comparing our paper with the work by Watanabe and Gotoh (henceforth WG) is a mistake on the following counts:

- WG studies polymeric turbulence (PHIT) while we study elastic turbulence (ET). We have now cited WG in the list of papers we cite about simulations of PHIT. The two phenomena shares the same equations but the parameter ranges are completely different as we have explained already in the introductory section of our paper and also illustrated in Fig. 1.
- WG studies a decaying problem while we study a forced one.
- WG does not consider the subdominant term in structure function. There was no need for them to do that because they were studying PHIT not ET.
- WG did not study intermittency.

3. **Reviewer 1** How are the Taylor scale Reynolds number and Kolmogorov scale defined? Are they defined for each ET case or based on the Newtonian counterpart?

Response: They are based on the Newtonian counterpart, but they must be interpreted with care in ET. We have now added the following text to the paper:

“Note an important difference between ET and usual Newtonian HIT. In the latter the Kolmogorov length (or time) scale is defined as the scale where the inertial and viscous effects balance each other. Although we continue to use the same definition – Re_λ and τ_K are calculated from the Newtonian DNS – these scales lose their usual meaning because ET appears at small Re at scales where the inertial term is negligible. The η we obtain is, as expected, quite close to the scale of energy injection. Henceforth we use the box-size L which is also the scale of energy injection, as our characteristic length scale.”

4. **Reviewer 1** FIG. 2 is not mentioned in the main text.

Response: Our apologies. We have now added the following text after Eq. (3).

“In Fig. 2 we show typical snapshots of the two energy dissipation rates on two dimensional slices of our three dimensional DNS. ”

5. **Reviewer 1** Watanabe and Gotoh (PoF, 2014) reported that the energy spectrum in decaying isotropic turbulence, with a comparable Reynolds number used in this study, obeyed the power law scaling $E(k) \sim k^{-\alpha}$. However, they found that α increases with the Weissenberg number. In this study, the scaling exponent α seems universal. This difference needs some further discussion.

Response: As we have mentioned already, WG studied a different problem. They study decaying PHIT while we study forced ET, thus, the difference is expected. Furthermore, WG finds scaling over a little less than a decade where the scaling exponent changes by a very small amount (4.1 to 4.6). We have a larger resolution (1024^3 vs 128^3) and our spectra is averaged over several snapshots in the statistically stationary state of turbulence, while the spectra of WG is calculated from one snapshot of a decaying simulation.

6. **Reviewer 1** From the manuscript, I am not convinced by the authors that it is necessary to use the second difference of the velocity to study the intermittency. In Fig. S3(b), the fourth and sixth order velocity structure functions already departure from the smooth scaling, i.e., $S_p \sim r^p$. The scaling exponents of S_p , the PDF and kurtosis of the velocity increment should be presented before the use of the second difference of the velocity.

Response: The Referee seems to be particularly confused about our use of a non-conventional structure function. We have now rewritten this part of our paper to remove any confusion, see section IIIA and IIIB of the present draft. We have also added the usual structure function S_p , and the PDF of of usual velocity increments to the Supplementary Material. The key message that comes out of this analysis is: Intermittency is a fundamental property of velocity increments but it is necessary to use the second difference to properly reveal the nature of intermittency and to reliably extract the exponents ζ_p .

7. **Reviewer 1** This paper presents an unconventional definition of structure function that utilizes the second difference of velocity to extract the scaling exponents. However, the paper lacks comparisons or discussions with widely-known Newtonian baselines in the field. It would be valuable to compare the results with existing theories or experimental observations to provide a broader context for the findings.

Response: We believe that this is not the case. Indeed,

- There is no widely-known Newtonian baseline here, because there is no Newtonian counterpart of ET.
- If the referee implies comparison with intermittency in homogeneous and isotropic *Newtonian* turbulence, we already did that in the previous draft. See e.g.,

– We quote from page 4 of the previous draft:

To understand the importance of this result, let us revisit the Kolmogorov theory of turbulence: In the limit $r \rightarrow 0$ at a finite viscosity μ_f , $\delta u_\alpha(\mathbf{x}, \mathbf{r}) \sim r$ since velocity gradients are finite. But if we first take the limit $\nu \rightarrow 0$ and then $r \rightarrow 0$ ($\nu \equiv \mu_f/\rho_f$ is the kinematic viscosity) $\delta u_\alpha(\mathbf{x}, \mathbf{r}) \sim r^h$ with $h \approx 1/3$. The velocity field is *rough*. In contrast, ET is, by definition, a phenomenon at a finite viscosity (small Reynolds number), thus, the limit $\nu \rightarrow 0$ does not make sense – the velocity field is always smooth. But the non-trivial nature of ET manifests itself in the first subleading term in the expansion:

$$\langle \delta u_\alpha(\mathbf{r}) \rangle \equiv \langle u_\alpha(\mathbf{x} + \mathbf{r}) - u_\alpha(\mathbf{x}) \rangle \sim G_{\alpha\beta} r_\beta + H_{\alpha\beta} r_\beta^h + \text{h.o.t.}, \quad (1)$$

where $G_{\alpha\beta}$ and $H_{\alpha\beta}$ are (undetermined) expansion coefficients, and $h \approx \zeta_2/2 = 3/2$ and this is brought out not by the velocity differences, but by the *second difference* of velocity across a length scale. This is the first important result of our work.

– Figure (4c) where we compared the multifractal spectrum of ET with the multifractal spectrum obtained in HIT by Meneveau & Sreenivasan.

Anyway, we have now added several other comparisons in the present version. And have also added two paragraphs comparing HIT and ET in section IV.

8. **Reviewer 1** Is there a critical scale for the polymers to affect elastic turbulence, or can it be inferred from the kinetic energy spectrum? Do the results in this paper imply that we should resolve scales much smaller than the Kolmogorov scale in ET?

Response: Yes, in ET we must resolve scales much smaller than Kolmogorov scale because the phenomena appears at precisely those scales. This is well-known since the discovery of ET. Actually the Kolmogorov scale loses its usual meaning as the small scale cutoff of scaling. See our response to item 4 above. The question of critical scale is a crucial one. In short, there is a critical scale in S_p but this scale, cannot depend on Re, and seems to be almost independent of De. See the discussion in Section IIIA of the main text and Supplementary Material sections IIA and IIC.

9. **Reviewer 1** The authors should provide more details about the simulations, such as how the time step of about $10^{-5}\tau_K$ was chosen and the number of snapshots for data analysis.

Response: In Newtonian flows the timestep is often estimated through the Kolmogorov time scale τ_K . This does not work in ET, we write the timestep in units of τ_K to illustrate this. Thus we choose the timescale such

that the dissipative terms are properly resolved. Timestep is small because we are solving explicitly the viscous term and viscosity is large. We use 18 time snapshots with successive ones separated by $\approx 1.6 \times 10^4 \tau_K$ in our analysis.

10. **Reviewer 1** In section III. RESULTS, the authors argue that the scaling in the kinetic energy spectrum and the polymer spectrum follows $\xi = 2\chi + 1$. It would be helpful to explain the physical meaning behind this relationship.

Response: As we mention in the paper, this can be obtained from a scaling argument. The crucial physical ingredient in the argument is the fact that in ET at small scales the viscous stress in the momentum equation must balance the polymeric stress. We have now provided the details of this scaling argument in Section IB of the Supplementary Material.

11. **Reviewer 1** It is difficult for me to understand the statement "... $\zeta_2 = 3$. This is consistent with $E(k)$ we obtained from our simulations." Can the authors provide further explanation?

Response: We have now completely rewritten this part of the text for improved clarity. See section IIIA of the main text. We hope this is clear now.

12. **Reviewer 1** In figure 3(d), it seems that there two scaling ranges for $r > \eta$ and $r < \eta$. There should be more discussions on this.

Response: This is related to the question of scales. In short, there is no scaling for $r > \eta$. In ET, the Kolmogorov scale does not have its usual meaning. We have now explained this in the last paragraph of section II of the main text. We quote below:

"Note an important difference between ET and usual Newtonian HIT. In the latter the Kolmogorov length (or time) scale is defined as the scale where the inertial and viscous effects balance each other. Although we continue to use the same definition – Re_λ and τ_K are calculated from the Newtonian DNS – these scales lose their usual meaning because ET appears at small Re at scales where the inertial term is negligible. The η we obtain is, as expected, quite close to the scale of energy injection. Henceforth we use the box-size L which is also the scale of energy injection, as our characteristic length scale."

13. **Reviewer 1** The definition of $G_{\alpha\beta}$ and $H_{\alpha\beta}$ in equation (9) requires further explanation.

Response: We have now rewritten this part, see item 7 above.

14. **Reviewer 1** In the supplemental material: II. Fluxes, the definitions of (S1a) and (S1b) appear to be opposite.

Response: Apologies, now it is corrected.

15. **Reviewer 1** Most figures only show the results for $De = 1, 3,$ and 9 . It would be beneficial to include HIT results for comparison. For example, in the supplemental material: III., the authors argue that the initial flat range disappears completely at $p = 6$ for $De = 9$. More data should be presented here to determine whether this phenomenon arises from elastic effects or high-order numerical errors.

Response: The referee maybe misunderstood that that $De = 0$ in this problem corresponds to HIT, but it does not. As we have mentioned before, the comparison with HIT must be done with care as the two phenomena are very different. Nevertheless, we had already compared our results with HIT wherever relevant. There are results in HIT that are similar to what we show in III, Fig. S3. This is in Fig 4 of Ref. [1] which was already cited in the main text of our paper. This disappearance of flat range has nothing to do with numerical errors. Furthermore, we have confirmed that all the phenomena we observe are results of purely elastic effects. There are many checks and balances we have applied (section I of Supplementary Material) which rules out numerical artifacts.

16. **Reviewer 1** The format of the references does not meet Nature Communication's requirements. For example: Physical Review Letters sometimes appears as "Physical review letters", some other times appears as "Phys. Rev. Lett.". Some of the authors' first names are abbreviated, some are not.

Response: Nature Communication's guideline for authors mention that such details are not important at the stage of submission. The paper goes through a detailed copy-editing phase once it is accepted.

Reviewer 3

I read the paper with interest and curiosity. It is well written and the results are novel and interesting. In the following I have two comments which the authors should take into account for the paper to be published.

1. **Reviewer 3** It is said in the paper that TDR (turbulent drag reduction) is observed in PHIT situation (large De and large Re). Actually, for pipe flow TDR is observed somehow in the intermediate region between PHIT and HIT. It is worthwhile to recall that finite extension of polymers, usually taken into account in the FENE-P models, is crucial to observe the Maximum Drag Reduction Asymptote (MDR) by increasing the polymer concentration. In Oldroyd model concentration plays no role and MDR cannot be observed. I suggest the authors to clarify this point to avoid confusion in the present literature. After all, their research has nothing to do with TDR.

Response: As correctly pointed out by the referee, this work has nothing to do with TDR. Hence, we have now removed the reference to TDR while discussing PHIT. We now mention TDR only once, just for historical reasons.

2. **Reviewer 3** From a more technical point of view, denoting by $\delta v(r)$ the difference $v(x+r) - v(x)$, then the quantity discussed by the author is $\delta v(r) + \delta v(-r)$. For the second order structure function we have:

$$\Sigma_2(r) = \langle [\delta v(r) + \delta v(-r)]^2 \rangle = 2S_2(r) + 2\langle \delta v(r)\delta v(-r) \rangle \quad (2)$$

The quantity $\Sigma_2(r)$ is the superposition of regular scaling r^2 coming from the first contribution and something else. The first term of the above expression goes as r^2 as shown by the author in the insert of figure (3c). So the intermittent scaling should be given by the second term (at least for small r).

It could be useful to check that this is the case. In other words I suggest that the author explicitly show that $\langle \delta v(r)\delta v(-r) \rangle$ displays anomalous scaling. This can be easily done by showing $\Sigma_2(r) - 2S_2(r)$. Eventually they should explain why this is not possible.

Response: We thank the referee for this very interesting question. Let us define $C(r) \equiv \langle \delta v(r)\delta v(-r) \rangle$. Then a consequence of our results is that:

$$2C(r) = \Sigma_2(r) - 2S_2(r) \sim Ar^3 - Br^2 \quad (3)$$

where A and B are two constants. Hence in the limit $r \rightarrow 0$, $C(r) \sim r^2$. In other words, $C(r)$ will not reveal the intermittent contribution which is the sub-dominant contribution. Rather $\Sigma_2(r)$ is constructed such that the dominant contribution of $S_2(r)$ and $C(r)$ cancels and the sub-dominant contribution is revealed. Below we plot the absolute value of $C(r)/r^2$ as a function of r on a log-lin scale for three Deborah numbers. At small r the plot becomes flat confirming our expectation. We have now discussed this in Section IIB of Supplementary Material.

Reviewer 4

This manuscript reports on intermittency in Elastic Turbulence (ET) in the low Reynolds number $Re \sim 1$, high Deborah number $De \gg 1$ limit. Since intermittency is regarded as a high Reynolds number phenomenon, this low- Re , ET-intermittency, although occurring sub-dominantly, appears surprising. The intermittency is quantified using spectra, probability density functions (PDF) of second differences of the velocity field and their moments, and the multi-fractal spectrum. The results seem to be cogently presented and the main result appears to be non-trivial. However, following are the issues I would like the authors to address.

1. **Reviewer 4** The flow is energized using an Arnold-Beltrami-Childress (ABC) forcing. How do the small-scale sub-leading intermittent statistics depend on the specifics of the forcing scheme? Can the authors comment on the generality of the results obtained?

Response: The reviewer asks how universal our results are. This is a valid but difficult question in the present context because – as we have already written in the introduction – the question of universality for even the velocity spectrum of ET is not yet settled. Furthermore, as the reviewer points out, in our case the universality must be checked for *sub-leading* terms. We have now used another forcing scheme [2] and calculated the exponents ζ_p for $p = 2$ and 4. Within errorbars they agree with the values we had earlier obtained – see Fig. 3(e) of the new draft. However, this does not mean that we have proven universality in general. Such a detailed work is left for a future publication.

2. **Reviewer 4** There appears no discussion of the Flow visualization Fig. 2 in the paper. It would be useful to connect Fig. 2 with the PDF discussion of Fig. 4 (a),(b).

Response: We apologize for our mistake and thank the referee for this excellent suggestion. We now mention Figure 2 twice. Once after Eq. 3 and then again while discussion Fig 4.

3. **Reviewer 4** The Kurtosis of the distributions of velocity second differences given in the Supplementary Material (SM) is shown to scale as $r^{-1.5}$. This corresponds to a more precipitous drop with increasing scale-size than seen in classical turbulence which the authors quote as $r^{-0.11}$ (the authors must include a more recent flatness number for classical turbulence). The authors may want to comment on the difference between the two exponents and its implications.

Response: We thank the referee for this comment. We have now discussed this in Section IIIB in the main text.

4. **Reviewer 4** In order for this interesting work to merit publication in Nature Communications the authors must surely expend greater effort into informing the reader about the importance and implications of this work.

Response: Taking the referee's comment in account we have now rewritten most of the paper. Particularly Section III(A), where we discuss in length the role of the subleading terms in velocity differences and section IV, the concluding section of the paper.

I believe that with suitable revisions along the suggested lines this manuscript can be considered for publication in Nature Communications.

-
- [1] Jörg Schumacher, Katepalli R Sreenivasan, and Victor Yakhot, "Asymptotic exponents from low-reynolds-number flows," *New Journal of Physics* **9**, 89 (2007).
 [2] V. Eswaran and S.B. Pope, *Computers and Fluids* **16**, 257 (1988).

REVIEWER COMMENTS

Reviewer #1 (Remarks to the Author):

In the revised manuscript, the authors had addressed most of my questions. The importance and implications of this work are now much clearer. Following my suggestion, the authors provide details on the use of the second difference of velocities in their definition of the structure function, which improves the readability of the paper. They also carried out additional numerical simulations using the FENE-P model which demonstrates the generality of their results.

However, there are still a number of issues that need to be addressed:

1. Although WG2014 investigated the effects of polymers in decaying homogeneous isotropic turbulence, the Reynolds number in their study ($3 \sim 10$) and in the present study (40) are in the same order. In addition, the dynamics is also dominated by the elastic stress of the polymers in the late stage of the WG study. A proper comparison between the two studies is needed. It should be noted that whether the turbulence is forced or not does not affect the small scale statistics which is the focus of both studies.

2. The authors investigated the intermittency in the dissipation range of ET with the methods of energy spectra, velocity structure functions, and the multifractal spectra of the fluid dissipation field. The authors plot the scaling exponents in Figures 3(a), 3(e), and 5(c). However, a detailed examination of the quality of the scaling and how they obtained the scaling exponents is lacking. Compensated plot and local slope are the golden standard to determine the quality and range of the scaling laws. I would suggest the authors to plot compensated energy spectra with vertical axis in linear scale, local slope of velocity structure functions, and the curves for ϵ_r to provide in the supplementary material for the completeness of the paper.

Reviewer #3 (Remarks to the Author):

The authors provide clear answers to questions of my previous review and I think that the paper can be published without further work.

Second referee report on NCOMMS-23-42570A “Intermittency in the *not-so-smooth* elastic turbulence”

The authors have made substantial improvements to this version of the manuscript. In particular, they have shown that ET-intermittency is not unique to one particular large-scale setup, thus broadening the range of flows where these results are applicable. The authors might want to address the following issues.

1. In equations (1b), (10a) the role of subscript α is unclear. The authors must either use the scalar dot product between vectors to denote the longitudinal projection or explicitly state that repeated indices implies summation.
2. The authors must quantify the error in the flatness exponents of -1.66 in figure 4.
3. The authors note in the revised manuscript that ET-turbulence is devoid of nonlinear effects of high Reynolds number Navier-Stokes turbulence where time reversal symmetry is broken by viscosity. A follow up to this work could be to examine the role of temporal symmetry in ET-turbulence.

Overall Comments

I believe that the revised manuscript should be published in nature communications.

Reviewer 1:

In the revised manuscript, the authors had addressed most of my questions. The importance and implications of this work are now much clearer. Following my suggestion, the authors provide details on the use of the second difference of velocities in their definition of the structure function, which improves the readability of the paper. They also carried out additional numerical simulations using the FENE-P model which demonstrates the generality of their results. However, there are still a number of issues that need to be addressed:

1. **Reviewer 1** Although WG2014 investigated the effects of polymers in decaying homogeneous isotropic turbulence, the Reynolds number in their study (3~10) and in the present study (40) are in the same order. In addition, the dynamics is also dominated by the elastic stress of the polymers in the late stage of the WG study. A proper comparison between the two studies is needed. It should be noted that whether the turbulence is forced or not does not affect the small scale statistics which is the focus of both studies.

Response: We have now commented on the results of WG2014 and how they compare with those in this work, especially when we discuss the energy spectrum in Section III of the main text alongside mentioning their results in the introduction of Section I.

2. **Reviewer 1** The authors investigated the intermittency in the dissipation range of ET with the methods of energy spectra, velocity structure functions, and the multifractal spectra of the fluid dissipation field. The authors plot the scaling exponents in Figures 3(a), 3(e), and 5(c). However, a detailed examination of the quality of the scaling and how they obtained the scaling exponents is lacking. Compensated plot and local slope are the golden standard to determine the quality and range of the scaling laws. I would suggest the authors to plot compensated energy spectra with vertical axis in linear scale, local slope of velocity structure functions, and the curves for ϵ_r to provide in the supplementary material for the completeness of the paper.

Response: We now provide the local slopes (and their deviations) of energy spectra and second-order second difference structure functions for all three polymer elasticities in the Supplementary Material. We have also shown the scale averaged energy dissipation $\langle \epsilon_r \rangle$ for when $De = 3$ and marked the range we use to compute the multifractal spectrum.

Reviewer 3

The authors provide clear answers to questions of my previous review and I think that the paper can be published without further work.

Response: We are grateful to the Reviewer for their endorsement of our work and recommending its publication. We thank the Reviewer once again.

Reviewer 4

The authors have made substantial improvements to this version of the manuscript. In particular, they have shown that ET-intermittency is not unique to one particular large-scale setup, thus broadening the range of flows where these results are applicable. The authors might want to address the following issues.

I believe that the revised manuscript should be published in nature communications.

1. **Reviewer 4** In equations (1b), (10a) the role of subscript α is unclear. The authors must either use the scalar dot product between vectors to denote the longitudinal projection or explicitly state that repeated indices implies summation.

Response: We thank the reviewer for pointing out this oversight on our part. The implicit summation over repeated indices is now clearly mentioned wherever necessary.

2. **Reviewer 4** The authors must quantify the error in the flatness exponents of -1.66 in figure 4.

Response: Following this suggestion, we now mention the mean exponents and their standard deviations for all three cases of polymer elasticity individually. This is now clearly mentioned in the caption of Fig.4 in the main text where now a comparison is made against a line of slope -1.60 rather than -1.66 .

3. **Reviewer 4** The authors note in the revised manuscript that ET-turbulence is devoid of nonlinear effects of high Reynolds number Navier-Stokes turbulence where time reversal symmetry is broken by viscosity. A follow up to this work could be to examine the role of temporal symmetry in ET-turbulence.

Response: Indeed, the point raised by the Reviewer is very interesting and we will investigate into it in the future as suggested. We thank the Reviewer for pointing us in this direction.

REVIEWERS' COMMENTS

Reviewer #1 (Remarks to the Author):

The authors have adequately addressed all my comments, now I would recommend a publication of the manuscript in NC.